# Double-tap gene drive uses iterative genome targeting to help overcome resistance alleles

Alena L. Bishop [ID][1], Víctor López Del Amo [ID][1], Emily M. Okamoto[1], Zsolt Bodai[2], Alexis C. Komor [ID][2] & Valentino M. Gantz [ID][1✉]

Homing CRISPR gene drives could aid in curbing the spread of vector-borne diseases and controlling crop pest and invasive species populations due to an inheritance rate that surpasses Mendelian laws. However, this technology suffers from resistance alleles formed when the drive-induced DNA break is repaired by error-prone pathways, which creates mutations that disrupt the gRNA recognition sequence and prevent further gene-drive propagation. Here, we attempt to counteract this by encoding additional gRNAs that target the most commonly generated resistance alleles into the gene drive, allowing a second opportunity at gene-drive conversion. Our presented "double-tap" strategy improved drive efficiency by recycling resistance alleles. The double-tap drive also efficiently spreads in caged populations, outperforming the control drive. Overall, this double-tap strategy can be readily implemented in any CRISPR-based gene drive to improve performance, and similar approaches could benefit other systems suffering from low HDR frequencies, such as mammalian cells or mouse germline transformations.

[1] Division of Biological Sciences, Section of Cell and Developmental Biology, University of California San Diego, La Jolla, CA 92093, USA. [2] Department of Chemistry and Biochemistry, University of California San Diego, La Jolla, CA 92093, USA. ✉email: vgantz@ucsd.edu

The rapid spread of homing CRISPR-based gene drives through populations can help curb the impact of vector-borne diseases worldwide[1–3]. For example, mosquitos can be modified with beneficial genes to prevent carried pathogens[4–6] or with detrimental gene alterations to suppress the vector population[7–9]. Gene drives also offer promising solutions in crop pest population control[10] and invasive species suppression[11,12], such as for the rodents[13,14] currently impacting island conservation efforts[15]. Briefly, CRISPR gene drives operate by biasing their own inheritance from Mendelian (~50%) toward super-Mendelian (>50%) by converting heterozygous germline cells to homozygosity. Gene-drive constructs encode both a Cas9 endonuclease and a guide RNA (gRNA) that targets the precise location where the gene-drive transgene is integrated into the genome. In a heterozygous individual, resulting from a gene-drive individual mating with a wild-type, the Cas9/gRNA complex cleaves the wild-type allele opposing the gene drive. The endogenous cell machinery repairs this double-stranded DNA break, which copies the drive element from the drive chromosome to the cleaved wild-type one[16,17]. When this process occurs in the germline of an individual, the inheritance is strongly biased towards the gene-drive transgene.

To repair the double-stranded DNA break, the germline has a bias towards the efficient and highly accurate homology-directed repair (HDR) pathway, which uses the intact strand—in this case, the strand containing the gene drive—as a template for repair. In some cases, however, alternative, error-prone DNA-repair pathways, such as non-homologous end-joining (NHEJ) and microhomology-mediated end-joining (MMEJ), can instead generate small insertions or deletions (indels) near the gRNA cleavage site, disrupting the gRNA recognition sequence and rendering these indels resistant to further cleavage[4,18,19]. Since such mutations can no longer be targeted by the drive and are passed onto the progeny, they can effectively counteract the spread of a gene drive through a population and obstruct field applications of these tools[20]. Additionally, when a gene drive is inherited from the mother, it has been shown that in both the fruit fly[19] and in *Anopheles* mosquitoes[4,5], the Cas9/gRNA complexes deposited in the egg can prematurely target the incoming wild-type male allele, before it can reach the proximity of the female chromosome, which would be used as a template for HDR. This dynamic leads to the early generation of indels that prevent further gene-drive conversion during later germline development. In extreme instances, the offspring of these animals will carry ~50% gene drive and ~50% resistance alleles[19], making this maternal effect a substantial and problematic source of resistance alleles as a gene-drive progresses within a population.

Previously, we built a trans-complementing gene drive (tGD) in *Drosophila melanogaster* (*D.mel*) that inserts a Cas9 transgene within the coding sequence of the *yellow* gene and a tandem-gRNA cassette at the *white* locus[19]. The gRNA transgene encodes two gRNAs, one targeting *yellow* (*y1-gRNA*) at the location where Cas9 is inserted, and the other targeting *white* (*w2-gRNA*) at the gRNA cassette insertion site. When the separate Cas9 and gRNA lines are crossed, the Cas9 protein can complex with the two gRNAs to cleave the wild-type *yellow* and *white* alleles, which leads to each of the transgenes being copied onto the opposing chromosome by HDR. While this action leads to super-Mendelian inheritance of both transgenes, we have also observed resistance alleles generated by the end-joining alternative repair pathways. In this previous work, we analyzed these resistance alleles by sequencing ~500 flies containing mutations at either the *yellow* or *white* locus, and observed that there were specific indels that appeared at a higher frequency than others, consistent with other findings in human cells[21–23].

We herein attempted to circumvent this phenomenon by supplementing a CRISPR-based homing gene drive with additional gRNAs targeting the most common resistance alleles generated by the drive process. This modification should provide a second opportunity for allelic conversion through HDR by allowing the drive element to also cut a subset of the resistance alleles, improving the overall gene-drive inheritance. To do this, we built the "double-tap" trans-complementing gene drive (DT-tGD) containing two extra gRNAs, one for *yellow* and one for *white*, each targeting one of the most prevalent resistance alleles formed at each locus by our original tGD(*y1,w2*)[19]. We test the DT-tGD system and show its ability to improve drive efficiency at both loci. We further show that the DT-tGD can specifically target the resistance alleles using the added gRNAs, and that this targeting results in efficient HDR conversion. Lastly, we show that the DT-tGD spreads more efficiently in caged populations than the tGD control, supporting its potential use for counteracting resistance alleles in field applications of this technology.

## Results

**Double-tap trans-complementing gene drive improves inheritance rates.** To evaluate whether an additional gRNA would improve inheritance by recycling indels generated by the primary gRNA, we designed double-tap versions (DT-tGD) of the previously tested tGD targeting the genes *yellow* and *white*[19]. Compared to tGD, this new arrangement includes two additional gRNAs within the construct inserted in the *white* gene (Fig. 1a). These additional gRNAs targets the most prevalent resistance alleles generated at either the *yellow* or *white* loci (*y1b* or *w2b*) by the primary gRNAs (*y1* or *w2*, respectively) (Fig. 1b)[19]. In the double-tap system, the primary gRNA (*y1* or *w2*) cuts first, and then, if a specific high-frequency indel is generated due to error-prone NHEJ or MMEJ repair, the secondary gRNA (*y1b* or *w2b*) can cleave the indel allele for another opportunity to copy the drive by HDR (Fig. 1a). Given that the most common indels identified in our previous analysis[19] only lack 1 base pair, the two secondary gRNAs designed here to target the indel at the same location do so with a length of 19 nt instead of the canonical 20 nt (Fig. 1b).

To test the DT-tGD system, we made three gRNA-constructs to be compared to the tGD(*y1,w2*) control (Fig. 1d). The control construct has two gRNAs, *y1* and *w2*, driven by *D.mel* U6-3 and U6-1 promoters, respectively, along with a GFP marker expressed in the eye to track the presence of the transgene phenotypically. The first construct, DT-tGD(*y1,w2,y1b*), carries a secondary gRNA for *yellow* (*y1b*) driven by the *Drosophila grimshawi* (*D.gri*) U6-C promoter (Fig. 1d). The second construct, DT-tGD(*y1,w2,w2b*), carries a secondary gRNA for *white* (*w2b*), also driven by the *D.gri*-U6-C promoter (Fig. 1d). The third construct, DT-tGD(*y1,w2,y1b,w2b*), carries both the secondary gRNAs (*y1b, w2b*) driven by *D.gri*-U6-A and *D.gri*-U6-C, respectively (Fig. 1d). These different U6 promoters were chosen due to previous success in a gene-drive setting and to avoid the problematic recombination that has been shown to occur within the gene-drive element if identical sequences are used[24]. All of these gRNA-constructs were then inserted at the same location of our tGD(*y1,w2*) control in the *white* locus and similarly marked with GFP so they could be combined with the same Cas9 line as the original tGD[19]. This line carries a Cas9 gene driven by the germline-specific *vasa* promoter, inserted in *yellow* at the *y1-gRNA* cut site and marked with DsRed expressed in the eye (Fig. 1d).

To test these three double-tap constructs, we performed genetic crosses to combine the two tGD components by mating Cas9-expressing males to gRNA-expressing females. From their

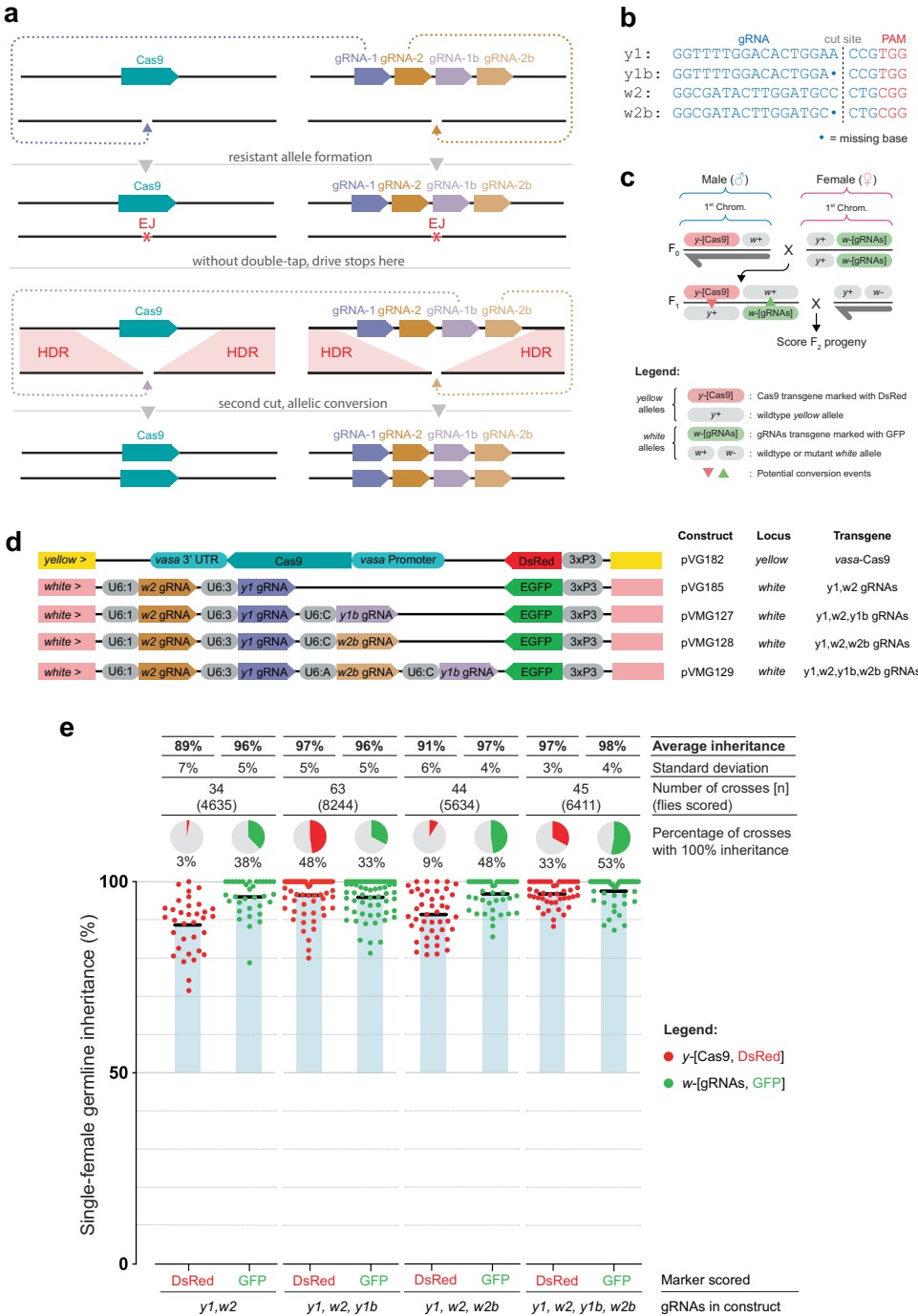

**Fig. 1 Double-tap trans-complementing gene-drive (DT-tGD) experimental setup and inheritance analysis. a** Schematic of the DT-tGD arrangement in which the Cas9 and gRNA elements are kept as two separate transgenic lines; gRNA-1 and gRNA-2 target the loci at which the Cas9 and gRNA elements are inserted, respectively. When crossed, Cas9 combines with gRNA-1 and gRNA-2 to generate double-strand breaks at each of the wild-type alleles. Repair by end-joining (EJ) pathways rather than homology-directed repair (HDR) would ordinarily halt gene-drive spread. Upon generation of a predicted resistance allele, Cas9 together with double-tap gRNAs gRNA-1b and gRNA-2b can regenerate double-strand breaks at these loci, providing a second chance for the drive elements to be copied by HDR. **b** gRNAs used in this system. The *y1*-gRNA and *w2*-gRNA target the wild-type *yellow* and *white* loci, respectively. The *y1b*-gRNA and *w2b*-gRNA target a single base pair deletion of the most common indel generated at the *yellow* and *white* loci, respectively. **c** Cross scheme used in this experiment. Males carrying the DsRed-marked Cas9 transgene inserted at the *yellow* locus are crossed to virgin females carrying the GFP-marked gRNA element inserted at the *white* locus. Trans-heterozygous virgin F$_1$ females are single-pair crossed to wild-type males, and the resulting progeny are scored for green and red fluorescence as markers of transgene inheritance. The dark gray half-arrows represent the male Y chromosome. **d** Transgenic fly lines used in this experiment. *vasa*-driven Cas9 is marked with DsRed and inserted in the *yellow* locus. Various gRNA combinations, in which each gRNA is driven by a U6 promoter, are marked with EGFP and inserted in the *white* locus. **e** Single female germline inheritance rates as measured by fluorescence phenotypes detected in the F$_2$ progeny. Black bars represent the average inheritance rates, and blue shaded boxes indicate the deviation from the normally expected 50% Mendelian inheritance. Pie charts represent the percentage of crosses that resulted in 100% inheritance of that transgene.

progeny, we then collected trans-heterozygous $F_1$ virgin females that should display gene-drive conversion in their germline and single-pair crossed them to wild-type (Oregon-R) males (Fig. 1c). We scored the $F_2$ progeny from each cross for the presence of the DsRed (Cas9) and GFP (gRNA) markers. Given that the *yellow* (Cas9-DsRed) and *white* (gRNA-GFP) are in close proximity (~1 cM) on the X chromosome, Mendelian inheritance would lead to ~50% DsRed and ~50% GFP F2 individuals; any individual carrying both elements signifies an allelic conversion event and a successful gene drive, with a small margin of error (~0.5%) due to meiotic recombination between the two loci.

For the tGD(*y1,w2*) control, we observed 89% inheritance of the Cas9-DsRed transgene and 96% inheritance of the gRNA-GFP transgene, in line with our previous characterization of this arrangement[19]. For the DT-tGD(*y1,w2,y1b*), Cas9-DsRed transgene inheritance improved significantly to 97% (compared to 89% for the control, $p < 0.0001$, Mann–Whitney test) (Supplementary Data 1), suggesting that the additional *y1b*-gRNA increases inheritance of the transgene. As expected, we did not see an increase in the inheritance of the gRNA-GFP transgene, which contained no secondary gRNA for this position and therefore displayed an average inheritance comparable with the control of 96%. For the DT-tGD(*y1,w2,w2b*), which instead carries an additional gRNA for *white*, we observed an average inheritance of 97% for the gRNA-GFP transgene (compared to 96% in the control). For this condition, the Cas9-DsRed transgene acted instead as an internal control displaying an average inheritance rate of 91% which is comparable with the control (89%). Lastly, we tested the four-gRNA DT-tGD(*y1,w2,y1b,w2b*), which we expected to improve the inheritance rates of both transgenes. Indeed we observed a significantly higher inheritance for the Cas9-DsRed in *yellow* (97% compared to 89% for the control, $p < 0.0001$, Mann–Whitney test) and the gRNA-GFP transgene in *white* (98% compared to 96% for the control, $p = 0.1179$, Mann–Whitney test) (Supplementary Data 1). From this analysis, we concluded that the double-tap arrangement can improve drive efficiency at the *yellow* locus. Given that the *w2*-gRNA has, on its own, very high conversion rates (~96%), we believe that the small range available for improvement did not allow us to observe statistically significant differences in these experiments.

We reasoned that the double-tap should also increase the overall number of crosses generating 100% inheritance due to its two-step action. Therefore, we compared the fraction of vials (i.e., germlines) producing 100% inheritance for each transgene. For DT-tGD(*y1,w2,y1b*), the fraction of vials producing 100% inheritance of the DsRed transgene climbed significantly from the tGD(*y1,w2*) control value of 3% to 48% ($p < 0.0001$, randomization test for a difference in proportions) (Supplementary Data 1). For DT-tGD(*y1,w2,w2b*), the fraction of vials displaying 100% GFP inheritance grew from 38% (control) to 48% with the double-tap ($p = 0.277$, randomization test for a difference in proportions) (Supplementary Data 1). Similarly for the four-gRNA DT-tGD(*y1,w2,y1b,w2b*), we observed a consistent increase in both transgenes, with the fraction of crosses at 100% DsRed inheritance significantly increased from 3% for the control to 33% ($p = 0.0006$, randomization test for a difference in proportions), and at 100% GFP inheritance increased from 38% for the control to 53% ($p = 0.133$, randomization test for a difference in proportions) (Supplementary Data 1). This additional analysis confirms that the double-tap can significantly improve inheritance at the *yellow* locus and, while all our observations are consistent with an improvement of inheritance at *white*, we did not observe statistical significance for these comparisons.

**Double-tap gene drive displays maternal effects caused by Cas9/gRNA deposition in the egg.** We then tested whether the double-tap drive would similarly improve inheritance when both the Cas9 and the gRNAs are co-inherited from the same parent, in a condition similar to a full gene drive[19]. To do this, we generated a homozygous fruit fly strain containing both the *vasa*-Cas9 and DT-tGD(*y1,w2,y1b,w2b*)-gRNAs on the same chromosome (Fig. 2). To first test the inheritance from a single parent without the additional confounding influence of maternal effects, we took males from this stock and mated them to wild-type virgin females. From the resulting progeny, we collected $F_1$ virgins and single-pair crossed them to wild-type males to evaluate the inheritance of the two transgenes (Fig. 2a). After scoring the $F_2$ offspring for the presence of the fluorophores, we observed 97% average inheritance of the Cas9-DsRed transgene and 98% average inheritance of the gRNA-GFP transgene, both significantly increased in comparison to the tGD(*y1,w2*) control (91% [$p < 0.0001$, Mann–Whitney test] for Cas9-DsRed and 96% [$p = 0.0093$, Mann–Whitney test] for gRNA-GFP) (Fig. 2c, Supplementary Data 2) and in line with our findings when the two elements were inherited separately from $F_0$ flies (Fig. 1e). When we analyzed the fraction of vials at 100% inheritance, we observed significant increases compared to the control for both the Cas9-DsRed transgene, from 6 to 38% ($p = 0.0003$, randomization test for a difference in proportions), and the gRNA-GFP transgene, from 27 to 49% ($p = 0.033$, randomization test for a difference in proportions) (Supplementary Data 2). These results suggest that a four-gRNA double-tap strategy significantly improves inheritance rates at both loci, further supporting the observation of the transgenes inherited separately from the $F_0$ and confirming the effect of the DT-tGD at the *white* locus described earlier (Fig. 1e). It is possible that this difference is due to the co-inheritance of the transgenes boosting the double-tap performance, or a statistical effect due to a higher number of crosses analyzed in the experiment in Fig. 2c.

The propagation of engineered gene-drive systems can suffer from a maternal effect caused by Cas9 protein and gRNA deposition in the egg by transgenic females, leading to the high-frequency generation of indels[4,5,19]. To evaluate if the double-tap system could alleviate this effect by recycling some of the generated indels, we crossed $F_0$ females from the Cas9+gRNA homozygous stock with wild-type males to obtain heterozygous $F_1$ females (Fig. 2b). We then single-pair crossed these $F_1$ females to wild-type males to evaluate the transmission of the two transgenes to the $F_2$ offspring (Fig. 2b). We observed that the Cas9-DsRed transgene was inherited at only 70% on average, similar to the tGD(*y1,w2*) control (Fig. 2c, Supplementary Data 2). The gRNA-GFP transgene displayed an even stronger maternal effect, with an inheritance rate of 49%, which was similar to the control (52%) and our previous findings[19] (Fig. 2c, Supplementary Data 2). These results suggest that the additional gRNAs in the double-tap arrangement are also deposited as Cas9/gRNA complexes in the egg and do not positively affect gene-drive performance through maternal inheritance. Our previous work showed that the primary gRNAs in this system (*y1* and *w2*) are extremely efficient and, when inherited by the mother, target the paternal allele in the first hours of development[19]. We speculate that the secondary gRNAs added to the double-tap arrangement are equally as efficient, given their similar sequences, and could therefore act in very rapid succession in the early stages of embryo development, effectively not overcoming the maternal effect.

**Double-tap secondary gRNAs specifically target indels for conversion.** To rule out an unexpected mechanism contributing

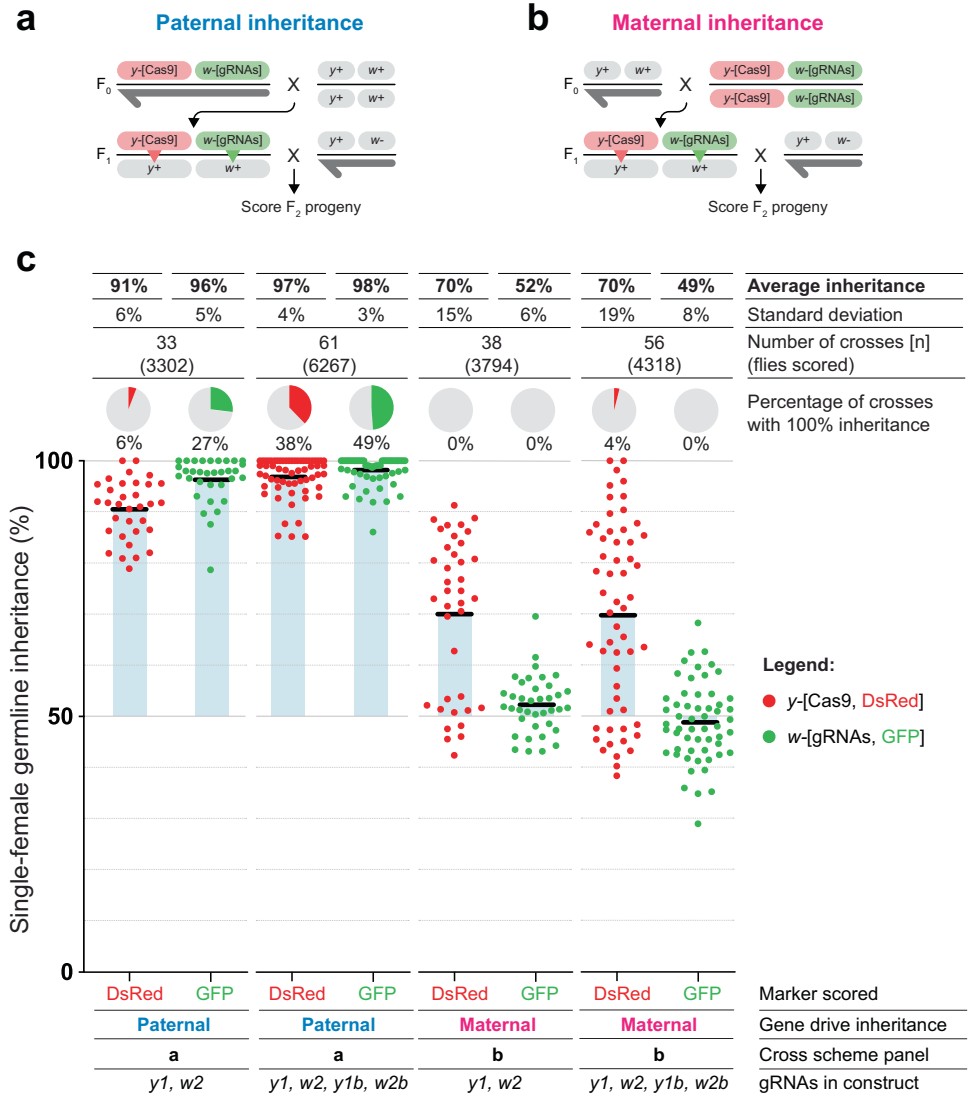

**Fig. 2 Maternal effect in the double-tap gene drive. a** Paternal inheritance cross scheme. $F_0$ males carrying both the DsRed-marked Cas9 element in *yellow* and the GFP-marked gRNA element in *white* are crossed to wild-type virgin females. Heterozygous $F_1$ virgin females are single-pair crossed to wild-type males, and $F_2$ flies are scored for red and green fluorescence as markers of transgene inheritance. **b** Maternal inheritance cross scheme. Homozygous $F_0$ females carrying both Cas9 and gRNA elements are crossed to wild-type males. $F_1$ cross and $F_2$ scoring are the same as in panel **a**. **c** Single female germline inheritance rates as measured by fluorescence detection in $F_2$ progeny. Black bars represent the average inheritance rates, and blue shaded boxes indicate the deviation from the normally expected 50% Mendelian inheritance. Pie charts represent the percentage of crosses that resulted in 100% inheritance of that transgene.

to the increased rate of transgene inheritance in the double-tap system, we evaluated the makeup of the indels and the prevalence of the *y1b* and *w2b* sequences in the $F_2$. To do this, we sequenced several DsRed- or GFP- $F_2$ males from the experiments performed in Fig. 1. We used males because they have only one X chromosome containing the *yellow* and *white* locus and therefore allow for sequencing of one copy of each of these loci. From each condition, we isolated several male flies and genotyped the indel generated at either the *yellow* or *white* locus. Indeed, the *y1b* sequence disappeared from conditions carrying the *y1b*-gRNA, while the *w2b* sequence disappeared from conditions carrying the *w2b*-gRNA (Table 1, Supplementary Fig. 1). These results suggest that the secondary gRNAs successfully target the intended indels, likely allowing for an additional round of gene-drive conversion.

To further show that the secondary gRNAs in the double-tap system specifically target the intended indels, we challenged the wild-type alleles with constructs lacking one of the primary

gRNAs (Fig. 3a). First, we generated two control tGDs (C-tGD), one containing *w2* and *y1b* gRNAs (without a *y1*) and one containing *y1* and *w2b*-gRNAs (without a *w2*); these constructs were otherwise the same as the tGDs described above and were inserted in *white* and marked with GFP (Fig. 3a). We then crossed $F_0$ C-tGD(*y1b,w2*) virgins to $F_0$ *vasa*-Cas9 males. To evaluate the transmission of the two transgenes, we collected trans-heterozygous $F_1$ virgins and outcrossed them to wild-type males in single pairs (Fig. 3b). Scoring the $F_2$ for DsRed and GFP expression, the gRNA-GFP transgene in *white* is inherited at super-Mendelian frequencies (96%) given the presence of the *w2*-gRNA, but the Cas9-DsRed transgene in *yellow* instead shows Mendelian inheritance (~50%), suggesting that the *y1b*-gRNA is unable to target the wild-type *yellow* sequence (Fig. 3d). To evaluate the *w2b*-gRNA in the same way, we then performed the same cross using the C-tGD(*y1,w2b*) (Fig. 3b'). Similarly, the Cas9-DsRed transgene was inherited at ~92% with the primary

**Table 1 Summary of the indel sequence analysis.**

| tGD gRNAs | Alleles observed / total (%) | | | |
|---|---|---|---|---|
| | **y1b** | | **w2b** | |
| y1,w2 | 18 / 37 (49%) | – | 12 / 19 (63%) | – |
| y1,w2,y1b | 0 / 28 (0%) | ****(p < 0.0001) | 13 / 27 (48%) | n.s. (p = 0.241) |
| y1,w2,w2b | 18 / 45 (40%) | n.s. (p = 0.291) | 0 / 38 (0%) | ****(p < 0.0001) |
| y1,w2,y1b,w2b | 0 / 27 (0%) | ****(p < 0.0001) | 0 / 21 (0%) | ****(p < 0.0001) |

p-values were calculated for the three experimental conditions in comparison with the control using a 1-tail randomization test for a difference in proportions.
n.s. not significant.
****p < 0.0001.

*y1*-gRNA, at about the same rate as the basic tGD(*y1,w2*) (Fig. 3d). The gRNA-GFP transgene instead showed Mendelian inheritance (~50%), suggesting that the *w2b*-gRNA is unable to cut the wild-type *white* allele (Fig. 3d). These experiments show that the two secondary gRNAs (*y1b* and *w2b*) are unable to target the respective wild-type sequences, at least not at a level detectable in this system.

We next wanted to demonstrate that the *y1b*- and *w2b*-gRNAs can specifically target the intended alleles to generate a gene drive via the conversion of these indels. To test this, we generated a fruit fly line termed "*y1b,w2b*", which carries the two indel alleles (*y1b, w2b*) generated at the respective loci by previous rounds of gene drive using the primary gRNAs. These alleles in this fruit fly line should be efficiently cleaved by the secondary gRNAs of the same name. We separately generated homozygous lines combining each of the C-tGDs with *vasa*-Cas9 on the same chromosome. We then crossed males from these *vasa*-Cas9,C-tGD stocks to *y1b,w2b* females; from their offspring, we collected $F_1$ heterozygous virgins and single-pair crossed them to wild-type males to evaluate the transgene transmission to their $F_2$ progeny (Fig. 3c-c'). For C-tGD(*y1b,w2*), we observe that *y1b*-gRNA can cut the *y1b* allele, leading to a super-Mendelian average inheritance of 93% of the Cas9-DsRed transgene, while the *w2*-gRNA, however, is unable to cleave the *w2b* allele, resulting in Mendelian inheritance of the gRNA-GFP transgene (52%) (Fig. 3e). Similarly, when we analyze the $F_2$ generation of the C-tGD(*y1,w2b*) cross, *w2b*-gRNA successfully triggers super-Mendelian inheritance of 91% of the gRNA-GFP transgene, while *y1*-gRNA does not seem to cut the *y1b* allele, leading to an observed Mendelian inheritance of the Cas9-DsRed transgene (51%) (Fig. 3e). Combined, these results show that each of the four gRNAs in our system specifically cleave the sequences they are meant to target, and all of them can generate a gene drive of the respective transgene.

**Double-tap improves drive when the number of gRNAs in the system is held constant.** Given that our DT-tGD carries four-gRNA-expressing genes while the control tGD(*y1,w2*) has only two, we tested whether differences in the total number of gRNA-expressing genes could affect gene-drive efficiency and therefore the interpretation of our double-tap results. Since the effect of the double-tap strategy is stronger on the transgene inserted in *yellow*, we focused on this transgene for this analysis. To control the number of gRNA genes, we generated an additional C-tGD carrying only two gRNAs, *y1* and *y1b*, analogous to the tGD(*y1,w2*) (Supplementary Fig. 2a). To comparably test these constructs where two gRNAs are expressed but only one locus is cut, we disabled the action of the gRNA targeting *white* using a version of our Cas9 line in which the *w2* cut site is destroyed by a 13 bp deletion that includes the PAM site (Supplementary Fig. 2b). With this Cas9,*w*$^{\Delta13}$ line, the *w2*-gRNA expressed by the tGD(*y1,w2*) construct can still bind to the available pool of Cas9, but it will not be able to cleave the genome at *white*. This makes

the *w2*-gRNA gene a placeholder in this system, allowing us to have the same number of gRNA-expressing genes across the two conditions without changing the number of cuts generated at one time.

To perform this experimental analysis, we took males from the Cas9, *w*$^{\Delta13}$ line, crossed them to virgins from either the tGD(*y1,w2*) or tGD(*y1,y1b*) lines, collected $F_1$ virgins, and crossed them to wild-type males to evaluate the inheritance of the respective constructs in the $F_2$ by scoring the fluorescent markers (Supplementary Fig. 2c-c'). As expected, the gRNA-GFP transgene inserted in *white* was inherited in a Mendelian fashion, given that the *w2*-gRNA is unable to cut the *w*$^{\Delta13}$ allele and that the tGD(*y1,y1b*) has gRNAs targeting only *yellow* (Supplementary Fig. 2d). In contrast, both conditions showed super-Mendelian inheritance at *yellow*, with the DsRed-Cas9 transgene in the tGD(*y1,w2*) present in an average of 91% of the $F_2$ flies and the C-tGD(*y1,y1b*) at a significantly higher average inheritance of 95% (*p* = 0.0018, unpaired t test) (Supplementary Fig. 2d, Supplementary Data 4). In addition, the percentage of crosses with 100% DsRed flies also increased from 0% with tGD(*y1,w2*) to 14% using DT-tGD(*y1,y1b*) (Supplementary Fig. 2d). These inheritance values are comparable to the previous experiments using tGD(*y1,w2*) and DT-tGD(*y1,w2,y1b,w2b*) (Fig. 1e), suggesting that the difference in gRNA-expressing constructs in our initial analysis is not responsible for the increase in inheritance observed for the DT-gRNA(*y1,w2,y1b,w2b*) construct. Together, these results confirm that the addition of secondary gRNAs to the double-tap system increases drive efficiency, which is not due to differences in the amount of gRNA-expressing genes in the double-tap transgene.

**DT-tGD outperforms regular tGD when spreading in a population.** Because the double-tap strategy improved gene-drive performance, we next wondered whether the addition of secondary gRNAs would improve the spread of the DT-tGD in a population. Given that our DT transgenes are inserted in either the *yellow* or *white* genes, to eliminate a fitness difference between the gene drive and the wild-type alleles we used a homozygous *yellow*-, *white*- fly line as our target population. For this purpose, we generated a mutant line in our OregonR laboratory background by injecting gRNA- and Cas9-expressing plasmids targeting the first exon of *yellow* and *white*. These null alleles, *y*$^{-EX1}$ and *w*$^{-EX1}$ were generated at a considerable distance from the gene-drive insertion site so as to not influence the sequence–homology-dependent gene-drive process (Fig. 4a).

To test the performance of the double-tap strategy in a caged population setting, we seeded three bottles with: (1) 50 *y*$^{-EX1}$, *w*$^{-EX1}$ virgin females; (2) 40 *y*$^{-EX1}$, *w*$^{-EX1}$ males; and (3) 10 males from a homozygous stock containing the *vasa*-Cas9-DsRed construct and either the tGD(*y1,w2*) control or the DT-gRNA(*y1,w2,y1b,w2b*) (Fig. 4b). These bottles, each containing 100 flies, were incubated at 25 °C and the parental generation was removed after 5 days. The next

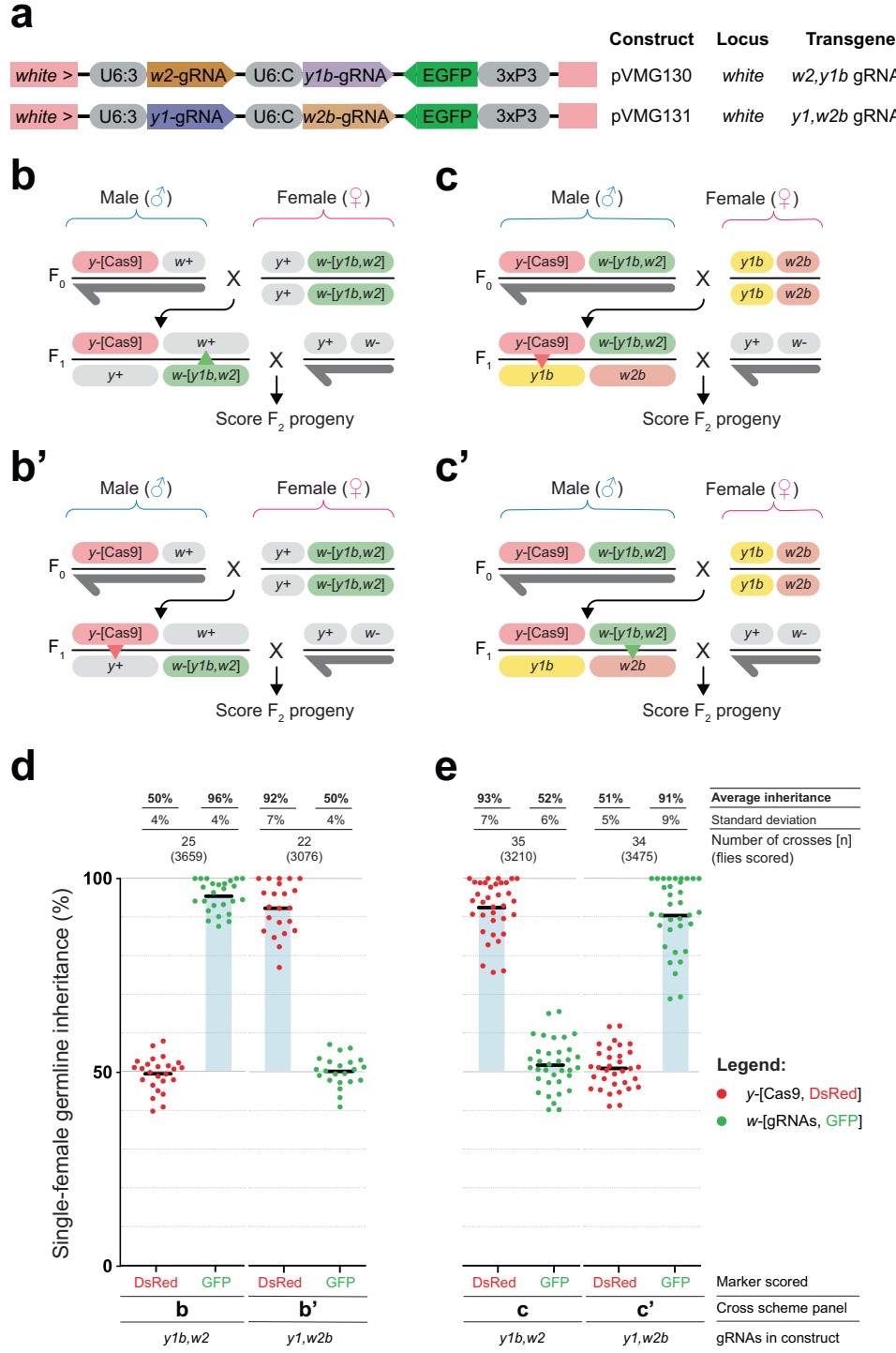

**Fig. 3 Specificity analysis of the gRNAs used in the double-tap system. a** Transgenic fly lines generated to test specificity of gRNAs. Different combinations of gRNAs driven by U6 promoters are marked with 3xP3-EGFP and inserted at the *white* locus—the same as all other gRNA lines used in this work. **b, b'** Cross scheme used for experiments in panel **d**. Males carrying DsRed-marked Cas9 inserted at the *yellow* locus are crossed to virgin females carrying one of the two GFP-marked gRNA elements inserted at the *white* locus. Trans-heterozygous $F_1$ virgin females are single-pair crossed to wild-type males, and the resulting $F_2$ progeny are scored for red and green fluorescence as markers of transgene inheritance. Symbols are the same as Fig. 1c. **c, c'** Cross scheme used for experiments in panel **e**. $F_0$ males carrying both the DsRed-marked Cas9 transgene and one of the two GFP-marked gRNA elements are crossed to virgin females homozygous for *y1b* (yellow box) and *w2b* (light brown box) alleles, which are single base pair deletions at each locus targetable by the *y1b*- and *w2b*-gRNAs, respectively. Heterozygous $F_1$ virgin females are crossed to wild-type males and the resulting $F_2$ progeny are scored for red and green fluorescence as markers of transgene inheritance. **d** Single female germline inheritance rates as measured by scoring fluorescence in $F_2$ progeny. Results from the **b, b'** crosses. Graph labeled as in Fig. 1e. **e** Same as **d** for the results obtained from the **c, c'** crosses.

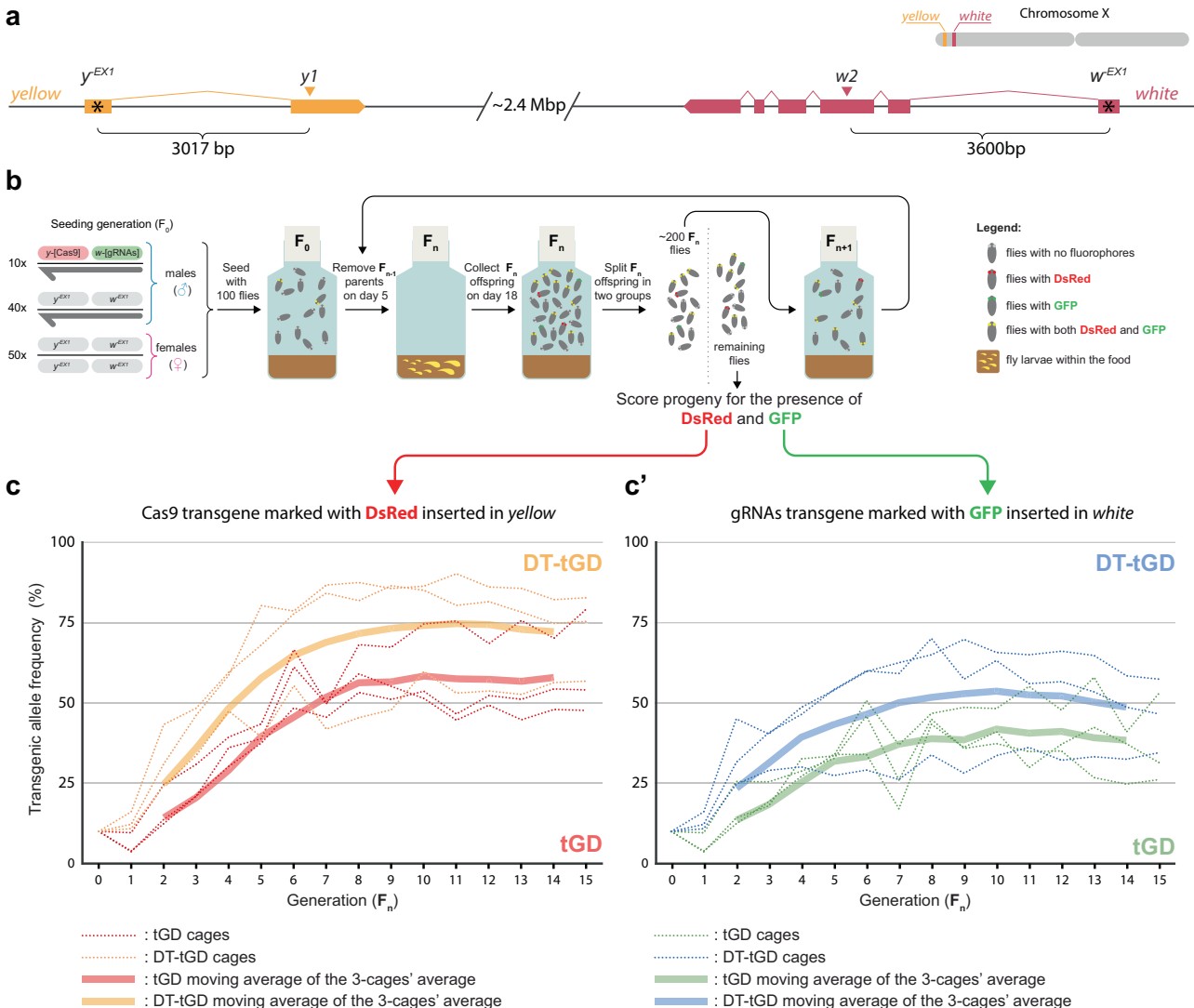

**Fig. 4 tGD(y1,w2) and DT-tGD(y1,w2,y1b,w2b) performance in caged populations. a** Schematic of the *yellow* and *white* genomic loci, indicating the locations targeted by the *y1*- and *w2*-gRNAs (triangles) and the *y^{-EX1}* and *w^{-EX1}* mutations (asterisks). An approximate location of *yellow* and *white* on the X chromosome is shown on the top right of the panel. **b** Schematic of population experiment. Cages are seeded with 100 flies, including 10 males that carry both the Cas9 and gRNA drive elements. After 5 days, the adult flies are discarded, and the larvae are allowed to develop. On day 18, flies are removed from the cage and split into two groups—~200 flies are randomly selected to seed the next generation; the remaining flies are scored for red and green fluorescence as markers of transgene inheritance. **c** DsRed-marked Cas9 and **c'** GFP-marked gRNA transgene prevalence in 3 independent populations per condition, tracked over 15 generations by scoring the two fluorescent markers. Dotted lines represent 3 independent cages. Fat solid lines represent the moving average of the 3 cages' average.

generation in the form of eggs and larvae was left to develop until day 18, when the hatched flies were collected for phenotypic scoring and for seeding the next generation (Fig. 4b). To track the spread of the transgene in each population, we scored a portion of the offspring for the presence of the GFP and DsRed transgene markers at each generation. Indeed, the frequency of the transgenic alleles in each bottle increased over time until stabilizing between generations $F_{10}$ and $F_{15}$ (Fig. 4c). On average, the DT-gRNA(y1,w2,y1b,w2b) had a higher prevalence of both the Cas9-DsRed (Fig. 4c) and the gRNA-GFP (Fig. 4c') transgene than our tGD(y1,w2) control, suggesting a positive effect of the secondary gRNAs.

In our caged population experiments, the percentage of transgenic alleles in each condition seemed to level off at different values much lower than 100%. Indeed, we expected this behavior given the strong maternal effect previously characterized at both loci[19]. This effect was more pronounced for *w2* than for *y1*, consistent with the observations described in Fig. 2c and the

lower values observed in the cage experiments for the gRNA-GFP transgene inserted in *white* (Fig. 4c'). Furthermore, while in our experimental setup gene-drive conversion only happens in females as both transgenes are located in the X chromosome, gene-drive arrangements targeting autosomal genes where conversion occurs in both sexes could further benefit from a double-tap approach.

To confirm this was due to maternal effects and simultaneously evaluate the generation of indels as the tGDs spread, we deep-sequenced the targeted loci from pools of male individuals, again for their simpler makeup of one allele per individual. We sampled three timepoints: during the initial exponential spread (F4), when the gene-drive spread began to slow (F8), and at the end of the experiment to evaluate the final population makeup (F15). As expected, the frequency of wild-type alleles diminished over time in all cages, reaching levels in the 0–26% range in the F15 generation, and indel alleles accumulated (Supplementary Fig. 3). We then analyzed the

frequency of either the *y1b* or the *w2b* sequence in these pools; the *y1b* allele appears as early as the F4 generation in the tGD(*y1,w2*) cages and seems to accumulate over time, with all tGD(*y1,w2*) cages containing it (Supplementary Fig. 3, Supplementary Data 6). Differently, in the DT-tGD(*y1,w2,y1b,w2b*), we observe the *y1b* allele only at low frequencies and only in two instances (F4, population 2; F8, population 1). These indels disappeared by the F15 generation, suggesting that when *y1b* alleles are generated and escape action of the *y1b*-gRNA, they can be targeted in subsequent generations (Supplementary Fig. 3, Supplementary Data 6). The *w2b* allele followed a similar trend, consistent with the elimination of the *w2b* alleles under the action of the *w2b*-gRNA present in the double-tap construct. Although here, we observed fairly high frequencies of the allele, but only in two out of three tGD(*y1,w2*) populations in the F15 (Supplementary Fig. 3, Supplementary Data 7).

Surprisingly, the *y1b* and *w2b* alleles accumulate in the tGD(*y1,w2*) populations at a much lower frequency than expected, given that in Table 1, we observe these alleles appearing with 49% (*y1b*) and 63% (*w2b*) in single-pair crosses. This may be explained by a qualitative difference between indel alleles generated through NHEJ/MMEJ in the late germline (Table 1) and indel alleles generated in population experiments. In the latter case, we expect that the major source of indel generation is the maternal effect which acts in early embryos, as seen in our previous study[19]. Altogether these results suggest that the double-tap strategy can improve gene-drive performance as it spreads in a population by specifically recycling indel alleles for the second round of gene-drive conversion.

## Discussion

Here, we developed the double-tap homing gene-drive strategy to combat the most prevalent resistance alleles that prevent drive spread. This strategy uses an additional, secondary gRNA targeting these resistance alleles to recycle them as new templates for an additional round of gene conversion, ultimately improving gene-drive efficiency. A double-tap version of a previously tested trans-complementing gene drive targeting the *yellow* and *white* loci of fruit flies[19] showed that the secondary gRNAs are specific in their targeting and improve the drive efficiency at both loci tested. The double-tap also improves the ability of the drive to spread in a population, with the double-tap reaching higher frequencies than the control.

Our work confirms that the efficiency of the drive depends on the locus and gRNAs used. Of the two loci tested here, the double-tap strategy performed better at *yellow*, likely due to the lower baseline conversion efficiency of *y1*-gRNA (89%) than *w2*-gRNA (96%). This generates more resistance alleles that can be further converted, which results in a more readily observable phenomenon for *yellow*. Additionally, this proof-of-principle work employed only one additional secondary gRNA, yet we still observed a modest improvement in efficiency. In the drive process, several resistance alleles are generated consistently, which could be targeted by the addition of multiple secondary or tertiary gRNAs to further improve conversion rates and approach 100% efficiency.

Our double-tap strategy also improves upon other proposed strategies that relied on the multiplexing of gRNAs to overcome resistance alleles. For example, two or more adjacent gRNA target sites have been employed to increase drive efficiency when either one of them would fail[25,26]. While this strategy allows for recycling resistance alleles, it also has the potential to generate non-homologous overhangs that can affect HDR rates, as we have shown in the previous work[19]. The double-tap acts instead as a multiplexing system "in time" instead of "in space" and creates no homology mismatches while still allowing the drive element

multiple chances to convert the wild-type allele. This feature of the double-tap system allows it to be seamlessly implemented in existing gene-drive systems to further boost their effectiveness.

Although this work addresses the drawback of indel formation slowing drive spread, another drawback of gene-drive systems in insects stems from the maternal effect caused by Cas9 and gRNA deposition in the egg, which severely impairs drive efficiency. While we hoped to improve on this issue, unfortunately, the double-tap does not seem to reduce this maternal effect, at least using the gRNAs tested in this study. We believe that the strong maternal effect observed here is due to the highly efficient gRNAs employed. Perhaps the use of less efficient gRNAs could lead to a lessened maternal effect and should also greatly benefit from a double-tap approach.

Finally, while the main scope of this work was to demonstrate the feasibility of the strategy, we also evaluated the constructs for their potential to spread in caged population experiments to test their potential for field use. While the strong maternal effect in both the double-tap and control populations rapidly generated resistance alleles that stifled the spread of either drive, we none-theless observed a higher level of spread for the double-tap than the control, further supporting the beneficial effect of the secondary gRNAs. This suggests that our double-tap strategy could be universally applied to increase the efficiency of CRISPR-based gene-drive systems suffering from resistance allele generation. For example, several mosquito systems[4,5,7,8] can partially circumvent the generation of resistance alleles by different strategies; implementing a double-tap approach should further increase their spread in a population. Additionally, secondary gRNAs could be used to specifically target problematic resistance alleles, such as those retaining target gene function and thus not suffering an imposed fitness disadvantage from the gene drive[5].

Finally, a double-tap strategy could be implemented in systems where HDR conversion is less efficient, such as primary human cells or mouse embryos. The use of secondary gRNAs in human cells could increase HDR-based transgenesis and perhaps benefit ther-apeutic uses requiring the HDR-based delivery of beneficial cargos[27], while its use in mice could further boost transgenesis efficiency beyond the latest improvements[28]. In fact in a complementary manuscript[29] we have applied the same double tap strategy to improve gene editing and transgene delivery efficiencies in human cells by using secondary gRNAs to recycle unwanted indel alleles for further rounds of editing. Overall, we expect the double-tap strategy to be widely applicable to diverse situations that could benefit from the use of secondary gRNAs to boost HDR efficiency or eliminate unwanted indels.

## Methods

All the work presented here followed procedures and protocols approved by the Institutional Biosafety Committee from University of California San Diego, com-plying with all relevant ethical regulations for animal testing and research. Gene-drive experiments were performed in a high-security Arthropod Containment Level 2 (ACL2) barrier facility.

**Plasmid construction**. All plasmids were cloned using standard molecular biology techniques. Plasmids were constructed by Gibson assembly using NEBuilder HiFi DNA Assembly Master Mix (New England BioLabs Cat. #E2621) and transformed into NEB 10-beta electrocompetent *E.coli* (New England BioLabs Cat. #3020). Plasmid DNA was prepared using a Qiagen Plasmid Midi Kit (Qiagen Cat. #12143) and sequences were confirmed by Sanger sequencing at Genewiz. Primers used for cloning can be found in Supplementary Information and the validated sequences of all constructs Have been deposited in the GenBank database; accession numbers are provided in the Supplementary Information and in the Data availability.

**Generation of transgenic lines**. Constructs were sent to Rainbow Transgenic Flies, Inc. for injection. All constructs were injected into our lab's isogenized Oregon-R (Or-R) strain to ensure consistent genetic background throughout experiments. Constructs were co-injected with a Cas9-expressing plasmid

(pBSHsp70-Cas9 was a gift from Melissa Harrison & Kate O'Connor-Giles & Jill Wildonger [Addgene plasmid #46294; http://n2t.net/addgene:46294; RRID: Addgene_46294]) and, if necessary, a pCFD3 plasmid (pCFD3-dU6:3gRNA was a gift from Simon Bullock [Addgene plasmid # 49410; http://n2t.net/addgene: 49410; RRID: Addgene_49410])[30] expressing previously validated gRNA-*w2*[31]. Injected $G_0$ animals were mailed back to us, then we outcrossed them to Or-R in small batches (3–5 males × 3–5 females) and screened the $G_1$ flies for a fluorescent marker (GFP expressed in the eyes), which was indicative of transgene insertion. We generated homozygous lines from single transformants by crossing to Or-R and identifying the white phenotype in subsequent generations. Stocks were sequenced by PCR and Sanger sequencing to ensure correct transgene insertion.

**Fly rearing and crosses**. All flies were kept on standard cornmeal food with a 12/12 h day/night cycle. Fly stocks were kept at 18 °C, and all experimental crosses were conducted at 25 °C. To phenotype and cross flies, they were anesthetized using $CO_2$. For all crosses, virgin females were crossed the same day that they eclosed. $F_0$ crosses were made in small batches of 3–5 virgin females crossed to 3–5 males. $F_1$ crosses were made in single pairs, left for 5 days, then the adults were removed. $F_2$ flies were counted as male or female and scored for the fluorescent marker (DsRed and/or GFP) using a Leica M165 F2 Stereomicroscope with fluorescence. We used DsRed or GFP expression as indicative of transgene inheritance. All gene-drive experiments were performed in a high-security ACL2 (Arthropod Containment Level 2) facility built for gene drive purposes in the Division of Biological Sciences at the University of California, San Diego. Crosses were made in shatter-proof polypropylene vials (Genesee Scientific Cat. #32-120) and all flies and vials were frozen for 48 h before being removed from the facility, autoclaved, and discarded as biohazardous waste.

**Sequencing of individual resistance alleles**. To sequence resistance alleles, we extracted genomic DNA from individual males following the protocol described by Gloor and colleagues[32]: flies were mashed in 50 μl squishing buffer (10 mM Tris-CI pH 8.2, 1 mM EDTA, 25 mM NaCI, and 200 μg/ml freshly diluted Proteinase K), then incubated at 37 °C for 30 min, then 95 °C for 2 min to inactivate the Proteinase K. We diluted each sample with 200 uL of water, then used 1–5 uL in a 25 uL PCR reaction spanning the gRNA cut site in either the *yellow* or *white* gene. The amplicon was then sequenced by Sanger sequencing to determine the resistance allele present. Primers used for resistance allele sequencing can be found in Supplementary Information.

**Caged population protocol**. For the population experiments, bottles were seeded with 100 flies each: (1) 50 $y^{EX1}$, $w^{EX1}$ virgin females; (2) 40 $y^{EX1}$, $w^{EX1}$ males; and (3) 10 males from a homozygous stock containing the *vasa*-Cas9-DsRed construct and either the tGD(*y1,w2*) control or the DT-gRNA(*y1,w2,y1b,w2b*). Each condition was performed in triplicate. Adult flies were left in the bottles for 5 days before being removed. The remaining eggs and larvae were allowed to develop until day 18 at which point all flies were anesthetized with $CO_2$, removed, and ~200 were chosen at random to seed the next generation. The remaining flies were phenotypically scored as male or female and for GFP and/or DsRed expression using a Leica M165 F2 Stereomicroscope with fluorescence, with the fluorescent markers being indicative of transgene inheritance. The bottles were maintained on this schedule for 15 generations. All experiments were done at 25 °C and flies were kept on standard cornmeal food with a 12/12 h day/night cycle. Experiments were conducted in shatter-proof polypropylene bottles (Genesee Scientific Cat #: 32-129 F) within the high-security ACL2 facility, maintaining the same precautions as previous other gene-drive experiments.

**Caged population deep-sequencing**. To perform deep-sequencing of the caged populations, we isolated 50 GFP-, DsRed- males from each cage at the generations F4, F8, and F15. For two samples we did not have 50 such flies available, and therefore we supplemented them with additional GFP-, DsRed+ flies (F8, Cage 2: 30 GFP-,DsRed- males and 12 GFP-, DsRed+ males; F8, Cage 3: 39 GFP-, DsRed- males and 11 GFP-, DsRed+ males). 50 OregonR WT males were used as an indel baseline control. Genomic DNA was extracted from each fly pool following the standard protocol in the DNeasy® Blood and Tissue Kit (Cat. No. 69504). After extraction, each sample was eluted with 300 uL of water, and about ~500 ng of the extracted DNA was then used in a 25 uL PCR reaction as a template to amplify either the *yellow* or *white* targeted region using specific primers for each locus (yellow F: ACACTCTTTCCCTACACGACGCTCTTCCGATCTCTCTGCTAATT CCGTATCCAGATTGGC, yellow R: TGGAGTTCAGACGTGTGCTCTTCCGAT CTGCCTATATCCACGGCAATGTTAGC, white F: ACACTCTTTCCCTACACG ACGCTCTTCCGATCTCTCTATTCGGCAGTCGGCCTGATCTG, white R: TGG AGTTCAGACGTGTGCTCTTCCGATCTGGTCATCCTGCTGGACATAGGC). One microliter of the resulting PCR reaction was used as a template for the subsequent PCR reaction to attach Illumina barcodes. Three microliters of the barcoding PCR product was then run on a gel, and the amplicon band was first gel extracted using QIAquick® Gel Extraction Kit (Cat. No. 28704) and then further purified using Monarch® PCR & DNA Cleanup Kit (5 μg) (Cat. No. T1030L). The pooled and purified DNA amplicons were quantified with the Qubit dsDNA high

sensitivity kit (Thermo Fisher). Equal amounts of amplicon from each sample were pooled together and prepared based on the Illumina sequencing protocol. 1.8pM of the pooled libraries were mixed with 1.8pM PhiX with nine to one ratio and loaded on an Illumnina MiniSeq instrument using a mid output kit of 300 cycles. Data were analyzed using CRISPResso2[33] to determine the frequency of resistance alleles across different generations; raw output data is summarized in Supplementary Data 6 and Supplementary Data 7.

**Caged populations data analysis**. Using as a reference the data obtained from the OregonR wild-type males we decided to consider any indel occurrence with less than 100 occurrences as background and removed these sequences from downstream analysis (removed sequences are in gray shading in Supplementary Data 6 and Supplementary Data 7). We then used the frequency observed for the different alleles (wild-type, *y1b* or *w2b*, and other indels are highlighted in yellow, red, and blue respectively in Supplementary Data 6 and Supplementary Data 7) to estimate the number of flies present in the sampled pool. The estimate was done by first dividing the frequency of a specific allele by the sum of all the frequencies of the alleles above the background (i.e., true alleles), then multiplying this number by the number of male flies that contributed an allele to the pool, and then rounding this number to the closest integer. The resulting estimates (i.e., number of flies contributing an allele to the pool) were used to generate the graphs in Supplementary Fig. 3.

**Graphical representation of the data and statistical analysis**. We used GraphPad Prism 9 and Adobe Illustrator to generate all the graphs. Statistical analyses were done using GraphPad Prism 9 and the StatKey analysis tool, version 2.1.1 (https://www.lock5stat.com/StatKey/index.html). For Fig. 1 and Fig. 2 we used a Kolmorgorov–Smirnov test to test for normal distribution and then Mann–Whitney tests to test for differences in means of inheritance rates. We also performed randomization tests for a difference in proportions to evaluate differences in the percentages of vials at 100% inheritance. In these analyses we performed 10,000 randomizations of our data. In Table 1 we again used randomizations tests for a difference in proportions with 10,000 randomizations to evaluate percentages of *y1b* and *w2b* alleles. For Supplementary Fig. 2 we performed a Kolmorgorov–Smirnov test to test for normal distribution and *t*-tests to evaluate the differences in means of inheritance rates.

**Reporting summary**. Further information on research design is available in the Nature Research Reporting Summary linked to this article.

## Data availability

The plasmid sequences of the constructs generated in this manuscript are either deposited into the GenBank database. GenBank accession numbers for the deposited plasmids are the following: pVG182 vasa-Cas9 (MN551085)[34], pVG185 tGD(y1,w2) (MN551090)[19], pVMG127 DT-tGD(y1,w2,y1b) (OL630771), pVMG128 DT-tGD(y1,w2,w2b) (OL630772), pVMG129 DT-tGD(y1,w2,y1b,w2b) (OL630773), pVMG130 C-tGD(w2,y1b) (OL630774), pVMG131 C-tGD(y1,w2b) (OL630775), pVMG138 C-tGD(y1,y1b) (OL630776); additional information is provided in the Supplementary Information. All source data are provided along with this manuscript. They cover the raw phenotypical scoring data collected in the gene-drive experiments, which are reported in the Supplementary Data 1-5 files, and the caged population experiment deep-sequencing data in the Supplementary Data 6-7 files all in Microsoft Excel format (.xlsx). All other data and information are available upon request from the authors.

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

## Acknowledgements

We thank Kaycie Butler for comments and edits on the manuscript. The research reported in this manuscript was supported by the University of California, San Diego, Department of Biological Sciences, by the Office of the Director of the National Institutes of Health under award number DP5OD023098 (to V.M.G.), by the National Institute of Allergy and Infectious Diseases under the award number R01AI162911 (to V.M.G.), by the National Science Foundation under the award number MCB-2048207.

## Author contributions

A.L.B. and V.M.G. conceived the project. A.L.B., V.L.D.A., E.O., Z.B., A.C.K., and V.M.G. contributed to the design of the experiments. A.L.B., V.L.D.A., E.O., Z.B., and V.M.G. performed the experiments and contributed to the collection and analysis of data. A.L.B. and V.M.G. wrote the manuscript. All authors edited the manuscript.

## Competing interests

V.M.G. is a founder of and has equity interests in Synbal, Inc. and Agragene, Inc., companies that may potentially benefit from the research results described in this manuscript. V.M.G. also serves on both the company's Scientific Advisory Board and the Board of Directors of Synbal, Inc. The terms of this arrangement have been reviewed and approved by the University of California, San Diego in accordance with its conflict of interest policies. A.C.K. is a member of the SAB of Pairwise Plants, and is an equity holder for Pairwise Plants and Beam Therapeutics. A.C.K.'s interests have been reviewed and approved by the University of California, San Diego in accordance with its conflict of interest policies. A.L.B., V.L.D.A., E.O., and Z.B. declare no competing interests.
