## [Peer Review File · Nature Communications]

Reviewers' Comments:

Reviewer #1:

Remarks to the Author:

Bishop et al investigate the benefit of the double-tap method for performing gene drives. The work is sound and the results very convincing, clearly showing that adding a second guide RNA cassette to target the most frequently generated resistance allele improves drive efficiency. Increased transmission of the gene drive alleles to progeny was demonstrated as well as increased propagation throughout a caged population. The generation of resistance alleles is a major challenge to gene drive propagation and the double-tap method appears to be a very useful option to alleviate its effect.

Minor comments

line 239, precise at which locus the indel sequences y1b and w2b were inserted.

Guide RNA efficiency seems to be the most important factor of drive efficiency. I wonder though, if the design of transgene insertion (exact insertion point relative to Cas9 cleavage site, transgene sequences abutting the chromosomal arms,...) also plays a role.

Large deletions are now a concern for gene editing experiments in mammalian cells. Have they been reported as possible resistance alleles in gene drives and have the authors examined whether they take place in their model system? Would their rates be increased when using two guide RNAs and two rounds of DNA cleavage in the double tap method?

I imagine that the authors were disappointed not to find that the double-tap method did not mitigate the maternal effect. Is it possible to reduce Cas9 or guide RNA expression levels/stability to minimize maternal deposition?

The text should mention the accompanying manuscript testing the double-tap method in human cell lines.

Reviewer #2:

Remarks to the Author:

Bishop et al describe a modification, referred to as "double-tap", that moderately improves the performance of homing-based CRISPR gene drives in the male germline of *Drosophila*. One of the barriers to successful gene drive has been the accumulation of NHEJ events at target loci. These events are resistant to subsequent Cas9 cleavage and thus block the spread of the gene drive. Even worse, NHEJ events that maintain the function of the associated gene can be subject to positive selection, to the detriment of the gene drive. Here, the authors characterize the most common NHEJ events recovered for gene drives targeting the yellow or white genes and added into their gene drive approach secondary guide RNAs capable of specifically cleaving these sequences. Overall, the experiments are robust and the authors interpret their results accurately and with caution where relevant.

Major findings:

In Figure 1, the authors document an increase in drive at yellow from 89% to 97%, with a greater number of fly batches exhibiting 100% inheritance. Improvement was not significant at white, which the authors attribute to the control already at 96%. Thus, the double-tap approach appears to be much more important for primary sgRNAs that are less efficient.

In Figure 2, the authors demonstrate that the benefits of the double-tap approach occur only in the male germline, maternally inherited Cas9/sgRNA still results in a predominance of NHEJ events. This reduces the impact of the double-tap method.

In Figure 3, the authors diligently determine that the double-tap benefit requires each sgRNA to cut in turn, and is not due to the second sgRNA cutting the wild-type allele or the 1st sgRNA cutting the mutant allele.

Finally, in Figure 4, the authors demonstrate that simple double-tap versions of a homing-based gene drive outperform the corresponding single sgRNA versions at both yellow and white target sites. While both gene drives fail to fully invade their target sites due to the accumulation of NHEJ events like their predecessors, this is nevertheless an important conceptual advance, since it is likely to make any homing approach more efficient.

Ultimately, this was a very well written manuscript with clear findings. There are no issues that I

can identify regarding experimental design, statistical analysis, or data interpretation. The only hesitation I might have is in regards to the significance of the double-tap approach. Since the initial description of the yellow gene drive, other groups have pursued different target genes and approaches (Crisanti group, James/Bier groups) that have demonstrated highly efficient gene drive that is not limited by the production of NHEJ events at their target sites. Thus, the problem the authors attempt to solve here appears to have already been solved to some extent. However despite this I think the double-tap approach has great potential in those cases where a limited number of predictable (and functional) alleles may arise and thus prevent their selection and thus can expand the number of targetable genes for various homing-based gene drive approaches.

Editorial comments:

Abstract and throughout:

"CRISPR-based gene drives" is too generic. The authors are referring to homing-based drives only, whereas CRISPR can also be used for gene drive architectures such as CLEAVR.

Introduction (paragraph two). "...which uses the existing strand..." Both strands exist, please rephrase.

Authors' response to the Reviewers

We thank both reviewers for their feedback on our manuscript, and for recognizing the value of the described double-tap approach and its potential use to boost the efficiency of homing CRISPR gene drives. The Reviewers' assessment helped us to improve the manuscript, and we hope to have addressed all the concerns raised and answered each point to satisfaction. We are providing a revised version of the manuscript, including the suggested edits and revisions, with comments highlighting the edits addressing each specific numbered comment. Please find here below point-by-point responses in blue text.

Reviewer #1 (Remarks to the Author):

1. Bishop et al investigate the benefit of the double-tap method for performing gene drives. The work is sound and the results very convincing, clearly showing that adding a second guide RNA cassette to target the most frequently generated resistance allele improves drive efficiency. Increased transmission of the gene drive alleles to progeny was demonstrated as well as increased propagation throughout a caged population. The generation of resistance alleles is a major challenge to gene drive propagation and the double-tap method appears to be a very useful option to alleviate its effect.

We thank the reviewer for appreciating our effort in presenting our findings in the manuscript, and underscoring the potential value of the double-tap strategy to alleviate the generation of resistant alleles and boost gene drive performance.

Minor comments

2. line 239, precise at which locus the indel sequences *y1b* and *w2b* were inserted.

Indeed as this sentence was written it was misleading, as the two *y1b* and *w2b* sequences were not inserted at a different locus; instead they are alleles generated by the primary gRNAs in previous rounds of gene drive using our control drive. We first isolated a *y1b* and a *w2b* alleles separately, and then combined them to generate a line homozygous for both alleles. We modified the text accordingly, to better clarify this point:

“To test this, we generated a fruit fly line termed “y1b,w2b”, which carries the two indel alleles (y1b, w2b) generated at the respective loci by previous rounds of gene drive using the primary gRNAs. These alleles in this fruit fly line should be efficiently cleaved by the secondary gRNAs of the same name.”

3. Guide RNA efficiency seems to be the most important factor of drive efficiency. I wonder though, if the design of transgene insertion (exact insertion point relative to Cas9 cleavage site, transgene sequences abutting the chromosomal arms,...) also plays a role.

Indeed we believe that the gRNA efficiency has a big impact on the gene drive efficiency (please see also our response to Reviewer's Comment #5). In this manuscript we have designed all our constructs to have perfect homology with the cleaved chromosome. Even the secondary gRNAs used here, which have a 1 bp deletion, target the same location and lead to homology arms that have perfect homology (even if 1 bp shorter). While not tested here, it would be possible to generate secondary gRNAs that target a small insertion. In such a case the homology arms generated would not have perfect homology with the transgene and this might affect HDR conversion; indeed, in previous work (López del Amo et al. 2020, <https://doi.org/10.1038/s41467-019-13977-7>) we show that even short mismatches in homology close to the Cas9 target site could have a detrimental impact on gene drive performance,

although HDR conversion is still occurring at an appreciable level. Separately, as the Reviewer suggests, the sequences within the transgene, abutting the homology arms could impact HDR conversion. In this manuscript we do not explore this aspect, although most of the constructs used have transgene sequences identical to each other in at least 300-1000 bp abutting the homology arms (Fig.1d, Supplementary Fig. 2a) suggesting that this is likely not a factor affecting our results. In Fig. 1 we are comparing transgenes that have different overall sizes which could on its own have an effect on HDR conversion. In our experience this is not likely as we have seen similar conversion rates for transgenes of different sizes inserted at the same location and driven by the same gRNA (also see López del Amo et al. 2020).

4. Large deletions are now a concern for gene editing experiments in mammalian cells. Have they been reported as possible resistance alleles in gene drives and have the authors examined whether they take place in their model system? Would their rates be increased when using two guide RNAs and two rounds of DNA cleavage in the double tap method?

In this work we have not pursued a thorough characterization of such alleles, and we acknowledge the fact that we did not identify such deletions here could be due to a bias given by the location of the primers used for detection. Since here we focused on the characterization of specific, small indels, which could be used for the double-tap strategy, we did not further analyze cases in which a PCR amplicon was not generated if the primers used fell within a large deletion.

In our experience we have observed large deletions happening during gene drive, although at a somewhat low occurrence. In previous work (López del Amo et al. 2020) we have performed a “primer-walk” approach to identify the break-points of such deletions, and observed very few of such large-deletion and large-insertion alleles. In other work (Xu et al. 2021, <https://doi.org/10.1016/j.molcel.2020.09.003>) we have observed larger deletions and also chromosomal rearrangements happening when both chromosomes are targeted at the same time. While these are important aspects to be aware of and to evaluate in the laboratory, gene drive applications aimed at release of gene-drive insects usually target insertion of the transgenes to essential genes or ones severely impacting viability. Nonetheless, it is hard to imagine a scenario in which large deletion of such important genes would lead to maintenance of these large-deletion alleles in the population. Therefore, the use of two or more gRNAs acting in succession, while it would increase the occurrence of large deletions, should not be a major concern for most field applications.

5. I imagine that the authors were disappointed not to find that the double-tap method did not mitigate the maternal effect. Is it possible to reduce Cas9 or guide RNA expression levels/stability to minimize maternal deposition?

Yes, indeed we were disappointed to see that a double-tap approach did not affect the maternal effect. Although in retrospect, this would have been a somewhat predictable result, given that the secondary gRNAs were also as efficient as the primary and would be deposited in the egg along with the primary ones.

Indeed, in a separate manuscript submitted to *Nature Communications* we show that the use of low-efficiency gRNAs can almost completely abolish the maternal effect by delaying cutting beyond early embryogenesis (see figure below). In this other manuscript we also show that low-efficiency gRNAs can produce gametes with wildtype alleles that are inherited untouched by the next generation. While here we use highly-efficient primary and secondary gRNAs which would both efficiently cut during early fruit fly development, our other work suggests that eventual indel alleles generated in a double-tap approach using low-efficiency gRNAs could be passed on to the next generation. This would lead to a double-tap strategy acting in a multi-generational fashion potentially further boosting the effect of a

double-tap approach. Indeed we believe that the combination of low-efficiency gRNAs and double-tab could lead to extremely powerful gene-drive systems. This is something that could be explored in the future although it is beyond the scope of this manuscript.

Figure. The low-efficiency *w5*-gRNA reduces the maternal effect. (a) Sequence of *w2* and *w5* gRNAs. Both gRNAs target overlapping sequences in the *white* gene. Each gRNA's CHOPCHOP score (a prediction of cutting efficiency) is listed alongside it. (b) Schematic of paternal inheritance. The gene drive elements are inherited from the male in the F_0 . F_1 females are crossed to WT males and the F_2 progeny are scored for transgene inheritance. (c) Schematic of maternal inheritance. The gene drive elements are inherited from the female in the F_0 . F_1 females are crossed to WT males and the F_2 progeny are scored for transgene inheritance. (d) Paternal and maternal inheritance of the *w2* drive resulting from b and c crosses. Single female germline inheritance was measured by scoring the GFP-marked gRNA cassette in the F_2 . Black bars represent average inheritance and blue-shaded boxes represent Super-Mendelian inheritance above the expected 50% line. (e) Paternal and maternal inheritance of the *w5* drive labeled and executed the same as d.

6. The text should mention the accompanying manuscript testing the double-tap method in human cell lines.

We have submitted two manuscripts to *Nature Communications*, one evaluating the double-tap in a gene-drive setting and one evaluating its potential use to boost HDR in human cells. Given that *Nature Communications* does not accept formal co-submissions, in the eventuality of acceptance of both stories we hope to coordinate with the editor the co-release of the two manuscripts, and in such an eventuality we will add text in both manuscripts referencing the other work.

Reviewer #2 (Remarks to the Author):

1. Bishop et al describe a modification, referred to as “double-tap”, that moderately improves the performance of homing-based CRISPR gene drives in the male germline of *Drosophila*. One of the barriers to successful gene drive has been the accumulation of NHEJ events at target loci. These events are resistant to subsequent Cas9 cleavage and thus block the spread of the gene drive. Even worse, NHEJ events that maintain the function of the associated gene can be subject to positive selection, to the detriment of the gene drive. Here, the authors characterize the most common NHEJ events recovered for gene drives targeting the yellow or white genes and added into their gene drive approach secondary guide RNAs capable of specifically cleaving these sequences. Overall, the experiments are robust and the authors interpret their results accurately and with caution where relevant.

We thank the reviewer for appreciating our work presented in the manuscript, and recognizing the potential that a double-tap approach could have not only to improve gene drive efficiency but also to target specific resistant alleles that maintain function of the gene.

Major findings:

2. In Figure 1, the authors document an increase in drive at yellow from 89% to 97%, with a greater number of fly batches exhibiting 100% inheritance. Improvement was not significant at white, which the authors attribute to the control already at 96%. Thus, the double-tap approach appears to be much more important for primary sgRNAs that are less efficient.

Indeed in this experiment we do not see a significant improvement at *white*, although we do see a significant improvement when a comparable experiment is performed in Fig. 2 (Paternal inheritance) supporting the observation in Fig. 1. Indeed we believe that the double-tap approach could be much more effective when applied to lower-efficiency gRNAs (see our response to Reviewer #1 Comment #5); Although we believe that there is an effect also with high efficiency gRNA such as *w2*.

3. In Figure 2, the authors demonstrate that the benefits of the double-tap approach occur only in the male germline, maternally inherited Cas9/sgRNA still results in a predominance of NHEJ events. This reduces the impact of the double-tap method.

Indeed, as also Reviewer #1 points out, we were hoping that this approach would help with the maternal effect generated by maternally inherited Cas9/sgRNA complexes. As we outlined in Reviewer #1 Comment #5 We believe that the combination of double-tap with strategies that lower maternal effect could further boost the impact that double-tap would have on the performance of a gene drive.

4. In Figure 3, the authors diligently determine that the double-tap benefit requires each sgRNA to cut in turn, and is not due to the second sgRNA cutting the wild-type allele or the 1st sgRNA cutting the mutant allele.

We thank the reviewer for this comment. Indeed we believed it was important to thoroughly evaluate this aspect. Given that the two primary and secondary gRNAs used here differed only by one nucleotide, they had the potential to act on each other's respective target.

5. Finally, in Figure 4, the authors demonstrate that simple double-tap versions of a homing-based gene drive outperform the corresponding single sgRNA versions at both yellow and white target sites. While both gene drives fail to fully invade their target sites due to the accumulation of NHEJ events like their predecessors, this is nevertheless an important conceptual advance, since it is likely to make any homing approach more efficient.

Indeed, here we aimed to compare the performance of the double-tap arrangement to the control. As such we eliminated any possible effect of fitness in the system by conducting the experiments in a w^- , y^- background. If combined with other strategies such as re-coding of the targeted genes we believe that these gene drive systems could have reached much higher prevalence in the cages.

6. Ultimately, this was a very well written manuscript with clear findings. There are no issues that I can identify regarding experimental design, statistical analysis, or data interpretation. The only hesitation I might have is in regards to the significance of the double-tap approach. Since the initial description of the yellow gene drive, other groups have pursued different target genes and approaches (Crisanti group, James/Bier groups) that have demonstrated highly efficient gene drive that is not limited by the production of NHEJ events at their target sites. Thus, the problem the authors attempt to solve here appears to have already been solved to some extent. However despite this I think the double-tap approach has great potential in those cases where a limited number of predictable (and functional) alleles may arise and thus prevent their selection and thus can expand the number of targetable genes for various homing-based gene drive approaches.

Yes indeed other ways have been devised to bypass the impact of NHEJ alleles on gene drive performance. One such approach is the introduction in the transgene of a re-coded portion of the gene disrupted by the gene drive. This approach gives the gene drive allele a higher fitness compared to any NHEJ allele disrupting gene function, which are weeded out the population by selection.

We see the double-tap approach being an upgrade and not a replacement of other strategies for two reasons: 1) a double-tap approach would recycle such NHEJ alleles for further conversion, producing more gene drive alleles coming out of a given germline; this would result of an increase of gene drive performance which even if low (e.g.: 5%) could lead to a great improvement when compounded over several generations. 2) A double-tap secondary gRNA could be designed to specifically target in-frame NHEJ alleles that retain gene function and would be otherwise detrimental to gene-drive spread as mentioned by the Reviewer in Comment #1.

Editorial comments:

Abstract and throughout:

7. “CRISPR-based gene drives” is too generic. The authors are referring to homing-based drives only, whereas CRISPR can also be used for gene drive architectures such as CLEAVR.

We have modified the abstract, introduction, and discussion to better clarify that this manuscript aims at implementing the double-tap in a homing CRISPR gene drive.

8. Introduction (paragraph two). “...which uses the existing strand...” Both strands exist, please re-phrase.

We modified the text changing “existing” to “intact”

Reviewers' Comments:

Reviewer #1:

Remarks to the Author:

The authors have satisfactorily addressed my minor concerns with the manuscript.

Reviewer #2:

Remarks to the Author:

All corrections have been made appropriately.